# Simulation Model Construction of Plant Height and Leaf Area Index Based on the Overground Weight of Greenhouse Tomato: Device Development and Application

Shenbo Guo [1,2], Letian Wu [1,3,4,*], Xinwei Cao [1,3,4], Xiaoli Sun [1,3,4], Yanfei Cao [2], Yuhan Li [2] and Huifeng Shi [1,3,4,*]

[1]  Institute of Agricultural Machinery, Xinjiang Academy of Agricultural Sciences, Urumqi 830091, China; guoshenbo01@163.com (S.G.)
[2]  College of Horticulture, Northwest A & F University, Xianyang 712100, China
[3]  Research Center for Agricultural Engineering Facilities and Equipment Engineering Technology, Urumqi 830091, China
[4]  Agricultural Engineering Company, Xinjiang Academy of Agricultural Sciences, Urumqi 830091, China
*  Correspondence: wuletian1982@126.com (L.W.); shihuifeng@sohu.com (H.S.)

**Abstract:** Plant height and leaf area index (LAI) are crucial growth indicators that reflect the growth status of tomatoes in greenhouses, enabling accurate determinations to effectively estimate crop transpiration and formulate irrigation strategies for reducing agricultural water waste. There is a need for the increased application of related models to simulate tomato growth indices in the traditional greenhouse production in China. This study proposes a nondestructive, real-time monitoring and simulation device for measuring tomato plant height and leaf area index. The weight of aboveground tomatoes was obtained by suspending tomato plants on dynamometers, while the total weight of stem and leaf organs was determined using a distribution coefficient simulation model. The $R^2$ value between the measurements from the electronic scale and those from the aboveground fresh weight device for tomatoes was 0.937, with an RMSE value of 0.05 kg. The monitoring device did not affect the average tomato growth during operation. The device will not affect the growth of tomatoes during monitoring. A multiple linear regression was used to compare the measured and simulated values of the plant height and leaf area index of various types of greenhouse tomatoes cultivated in different greenhouse types. The average $R^2$ value for simulating plant height was 0.817 with an RMSE of 10.81 cm. The average $R^2$ value for the leaf area index was 0.854, with an RMSE of 0.55 $m^2 \cdot m^{-2}$. The simulated values for plant height and leaf area index closely matched the measured values, indicating that the model has high accuracy and applicability in traditional Chinese greenhouses (solar greenhouses and insulated plastic greenhouses). However, further optimization is required for commercially produced, continuous plastic greenhouses equipped with greenhouse environmental control equipment.

**Keywords:** greenhouse; weighing device; tomato simulation model; multiple linear regression

## 1. Introduction

In recent years, protected horticulture has experienced remarkable growth, substantially increasing the cultivated area [1,2]. China boasts over 2.8 million hectares of horticultural facilities, representing over 80% of the world's total. This area includes 810,000 hectares (29%) of solariums and 1.52 million hectares (53%) of large- and medium-sized greenhouses [3]. These figures underscore the significant and growing role of facility agriculture in meeting the world's food needs, and the supply of vegetables is crucial while promoting sustainable development in facility agriculture [4]. Nevertheless, the expansion of facility agriculture also means a rise in water usage for agricultural purposes. As water scarcity becomes a growing problem worldwide, it challenges the sustainable development of facility agriculture [5,6]. By 2050, agricultural water scarcity is expected to spike in over

80% of the world's countries [7]. This leaves only a limited scope for increasing the water supply in agriculture, underscoring the need to enhance agricultural water use efficiency. The accurate calculation of the required crop water, also known as crop evapotranspiration (ETc), can help optimize irrigation management and augment water use efficiency during the growing season [8].

Tomatoes are a vital vegetable crop grown worldwide due to their easy cultivation and high economic efficiency [9–11]. To ensure that tomatoes grow and develop optimally, they must have an adequate water supply [12]. However, in traditional tomato production facilities such as solar and plastic greenhouses, irrigation is mainly based on artificial experience, which often results in significant water wastage [13]. The leaf area (LA) is the primary organ responsible for transpiration in tomato growth, and the leaf area index (LAI) is the total tomato leaf area per unit of land area, which is directly related to the crop evapotranspiration (ETc) of tomatoes [14]. Moreover, the plant's height is an essential indicator for assessing crop evapotranspiration (ETc) and transpiration in tomatoes, which is necessary for a high and stable tomato production [15,16]. Therefore, the real-time and accurate determination of the tomato's LAI and plant height is crucial for estimating crop evapotranspiration (ETc), which, in turn, provides a theoretical basis for evaluating the tomato water demand, making irrigation decisions, and reducing water wastage. Accurate estimates of crop evapotranspiration help greenhouse managers make irrigation decisions and promote the efficient use and conservation of resources [17].

Many experts and scholars have conducted a lot of research on the accurate determination of leaf area and plant height in tomatoes. Traditional leaf area measurements usually use destructive sampling methods to collect and determine the leaves, such as the punching and weighing method [18], Image J image processing method [19], leaf area meter method [20], etc. Although these type of methods can determine the leaf area more accurately, they destroy the normal growth of tomato plants, and the workload is large and time consuming. In order to determine the leaf area nondestructively, leaf area simulation has been studied by many scholars. L. Bacci et al. [21] improved the TOMGRO model to increase the simulation accuracy of leaf area; Wang X. et al. [22] developed a leaf area simulation model for tomato processing based on the physiological development time, using the physiological development time of tomatoes as the time scale; Wang D. et al. [23] developed a simulation model of pepper leaf area applicable to solar greenhouses using auxiliary heat product as a scale. Traditional plant height measurements are usually conducted directly using a tape measure, to save labor costs. Chang Yibo et al. [24] used a binary quadratic orthogonal rotated combinatorial design to establish a logistic model for tomato plant height based on radial heat product with an irrigation limit and fertilizer application as determinants; Cheng Chen et al. [25] constructed a celery plant height simulation model based on the relationship between greenhouse celery plant height and key meteorological factors (air temperature and solar radiation) with the single-plant irradiated heat product as the independent variable; Zhai Zihe et al. [26] used the XGBoost model to establish a simulation model for the plant height growth of cucumber at five fertility periods in a solar greenhouse, and the accuracy of the model was good. The LAI and plant height can be more accurately simulated through crop growth modeling, but the model requires the input of multiple environmental parameters, and the lack of costly IoT environmental monitoring equipment in most traditional greenhouses makes these types of models less practical.

Therefore, through the real-time determination of tomato leaf area and plant height, a scientific foundation was established to assess tomato crop water requirements and make irrigation decisions in conventional greenhouses. This study's mathematical simulation model of the tomato leaf area index and plant height was constructed using multiple linear regression based on experimental data from insulated plastic greenhouse fall stubble (the leaf area index was calculated from leaf area simulation results). The model was validated with experimental data from insulated plastic greenhouse spring stubble and solar greenhouse fall stubble. Additionally, a real-time monitoring device for the aboveground weight of tomatoes was designed to analyze the distribution coefficients of aboveground

stems and leaves among different stubbles. This information was used to establish a mathematical simulation model of the tomato leaf area index and plant height based on real-time fresh weight, which was further tested with experimental data from overwinter stubble in solar greenhouses and continuous plastic greenhouses. Furthermore, a low-cost and stable operation device for the real-time simulation of the leaf area index and plant height of tomatoes applicable to traditional greenhouses was developed and installed. The aim was to provide a tool for the nondestructive determination of individual or group tomato leaf areas and plant heights in traditional greenhouses while offering scientific references for estimating the evapotranspiration of tomato crops and water-saving irrigation.

## 2. Materials and Methods

### 2.1. Test Scheme

The experimental programmer involved three subtests to construct simulation models for the tomato leaf area index and plant height, a trial operation of a real-time monitoring device for tomatoes' aboveground fresh weight, and creating a simulation model for the tomato leaf area and plant height based on the real-time fresh weight data. Three different structural types of greenhouses were selected for the test greenhouses. Figure 1a presents a schematic of a large-span, asymmetric plastic greenhouse; Figure 1b presents a structural diagram of a solar greenhouse; Figure 1c presents a structural schematic of a continuous plastic greenhouse. The experimental crops involved medium- and large-fruited tomato varieties, and simulation models were conducted for the leaf area and plant height of tomatoes in the mid- and late-growth stages, specifically during the flowering and fruiting stages. The leaf area index was calculated based on the actual planted area after the leaf area simulation. The tomato plants were grown in substrate bags measuring 56 cm × 24 cm × 10 cm, with approximately 7 L of substrate per bag, and two tomato plants were planted in each substrate bag.

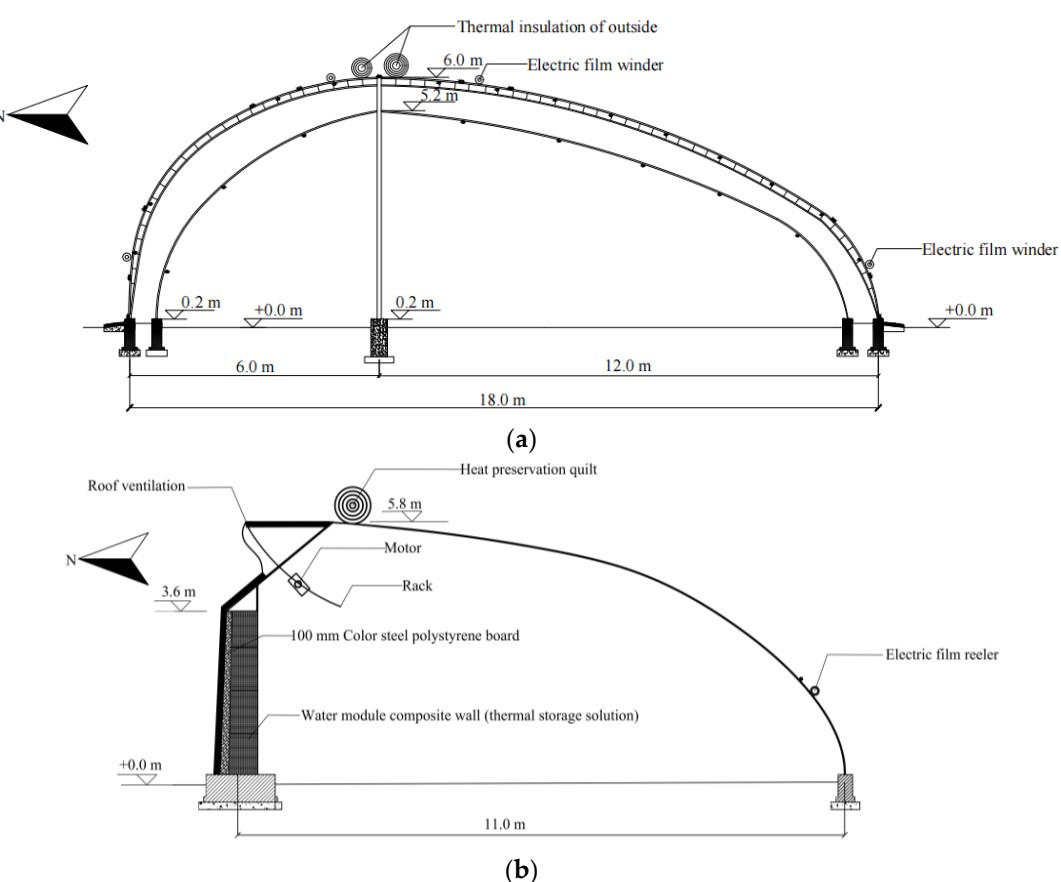

**Figure 1.** *Cont.*

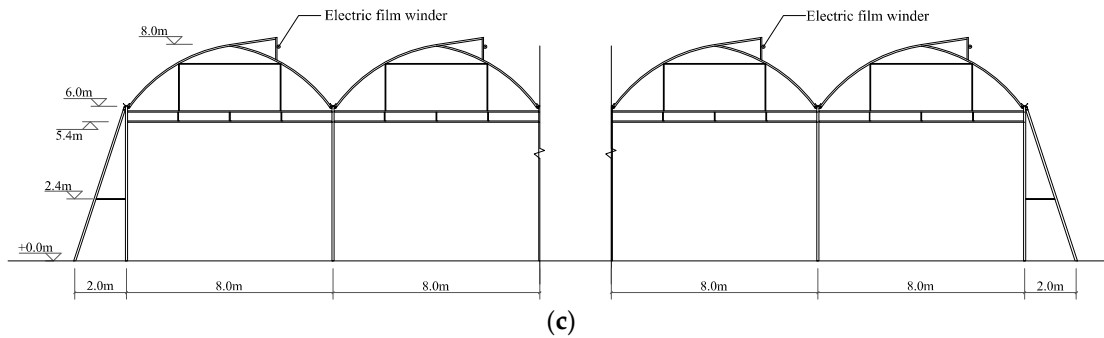

(**c**)

**Figure 1.** Three types of greenhouse structures. (**a**) Schematic of a large-span asymmetric plastic greenhouse. (**b**) Structural diagram of a solar greenhouse. (**c**) Structural schematic of continuous plastic greenhouse (partial).

2.1.1. Simulation Modeling Test for Leaf Area and Plant Height in Tomato

1. Cultivation experiment

Three stubble cultivation experiments were conducted at two locations: From January to May and from August to December 2021 in a double-arch, double-film, asymmetric, large-span insulated plastic greenhouse (Figure 1a) in the Modern Agricultural Integration and Experience Park in Yangling, Shaanxi, China (34°31′ N, 107°97′ E). The insulated plastic greenhouse was situated in a north–south direction, spanning 18.0 m with a ridge height of 6.0 m and an east–west length of 70.0 m. It had a double-layer skeleton. The inside and outside layers of the roofs on the north and south sides were covered with 0.12 mm plastic film, and the outside plastic film was covered with a 20 mm insulation quilt. Two roof vents and two bottom vents were placed on the north and south roofs for ventilation and cooling. The test tomato variety for the spring stubble was 'Provence', and the test tomato variety for the autumn stubble was 'Bao Lu Fu Qiang', with regular field management.

The experiment was conducted from January to May 2022 in a solar greenhouse (Figure 1b) at the horticultural farm of the North Campus of Northwest Agriculture and Forestry University (34°29′ N, 108°07′ E). The greenhouse had a span of 11.0 m, a ridge height of 5.8 m, and a length of 50 m in the east–west direction, and the tomato cultivar for the test was 'Cadjarli' with conventional field management.

2. Growth indicators and plant quality data acquisition

Three tomato plants were randomly selected for height measurement every seven days during tomato cultivation, and the distance between the top of the root and the apical growing point of the tomato was measured with a tape measure. Afterward, the three tomato plants were weighed using the destructive sampling method using a TCS-300 electronic scale (Jinhua, China) for the fresh weight of roots, stems (including petioles), flowers, and fruits, respectively. The leaf area of each tomato plant was determined using a CL-202 leaf area scanner (CID BioScience, Camas, WA, USA).

3. Simulation modeling of tomato leaf area and plant height

Autumn stubble data from asymmetrically insulated plastic greenhouses from August to December 2021 were selected for the construction of simulation models of tomato plant height and leaf area based on the following considerations:

The fresh weight of tomato plant stems as the independent variable and plant height as the dependent variable; The fresh weight of tomato plant leaves as the independent variable, and leaf area as the dependent variable.

Mathematical models of the linear function, polynomial function, and power function were established using the method of least squares regression, and the accuracies of the models were verified using spring stubble data.

4. Simulation of the aboveground morphological organ allocation index of tomatoes

Based on the actual tomato planting results in the cultivation experiment, combined with the conclusions of previous studies on tomatoes' aboveground morphological organ allocation index, the method of least squares regression was utilized to fit the change curves

of the allocation indices of the fresh weight of aboveground stem and leaf organs during the reproductive period of the tomatoes and to establish the calculation methods of the allocation indices of the fresh weight of aboveground stem and leaf organs of the tomatoes.

5. Model Validation

The root mean square error (RMSE) and mean absolute error (MAE) were used as indicators for the validation of the long-term monitoring accuracy of the device [26].

$$\text{RMSE} = \sqrt{\frac{1}{n}\sum\nolimits_{i=1}^{n}(ac_i - sm_i)^2} \tag{1}$$

$$\text{MAE} = \frac{1}{n}\sum\nolimits_{i=1}^{n}|ac_i - sm_i| \tag{2}$$

In this formula, $ac_i$ is the measured value; $sm_i$ is the simulated value; and $n$ is the number of samples.

2.1.2. Operational Testing of Real-Time Tomato Aboveground Weight Monitoring Device

1. Operational test

Experiment 1 was conducted from April to June 2022 in a solar greenhouse at the horticultural farm of the North Campus of Northwest A&F University of Science and Technology (NWAFU).

2. Device design

The weighing device consists of a data acquisition module, a communication module, and a monitoring module (Figure 2). The data acquisition module adopts an S-type tension sensor (DYLY-103, range: 0–10 kg, accuracy: 0.03%, Bengbu Dayang Sensing Company based in An Hui China). One end of the sensor is secured to trailing wire, and the tomato plants are suspended on the sensor using the trailing rope at an angle to conduct real-time measurements of the fresh weights of the tomatoes on the ground. The data are transmitted analogically in real time through a 4–20 ma current to a collection cabinet, the acquisition module (DAM3232, Bei Jing China Jiuying Soaring Electronics). The communication module adopts the 485 protocol to connect the serial port of the acquisition module and the 4G IOT module wired to transmit the data to the server, and the monitoring module adopts the 485 protocol to connect the serial port of the acquisition module and the MCGS touch screen wired to realize a local real-time display of the data.

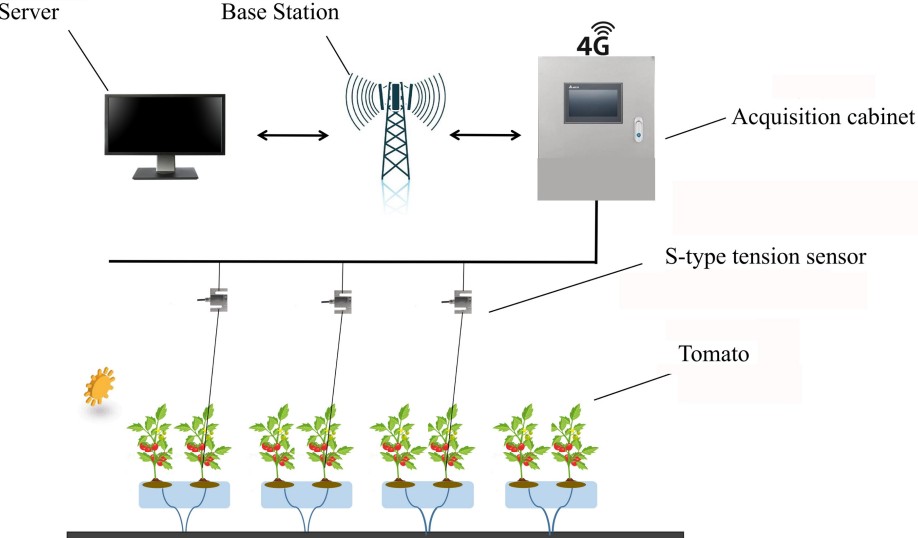

**Figure 2.** Schematic of real time-monitoring device for tomato aboveground weight.

3. Inspection of real-time tomato aboveground weight monitoring device

Long-term operation: From 23 May to 7 June 2022, 1.550 kg objects were suspended using three groups of S-type tension sensors. The collection interval was 1 h. Data were collected from the three sensor groups at the end of the inspection. The root mean square error (RMSE) and average absolute error (MAE) were used to verify the accuracy of the long-term monitoring device.

Measurement accuracy of weighing device: After measuring the aboveground weight of single tomato plants through the real-time tomato aboveground weight monitoring device, a destructive sampling method was used to weigh the intercepted tomato aboveground using an electronic balance, and the difference between the device measurement value and the actual measurement value was determined. A total of 45 tomato plants were selected for the accuracy test of the weighing device in the solar greenhouse on three occasions (with an interval of 20 days), and the root mean square error (RMSE) and mean absolute error (MAE) were used as indicators for verifying the long-term monitoring accuracy of the device. The root means square error (RMSE) and mean absolute error (MAE) were used to test the accuracy of the measurement.

### 2.1.3. Real-Time Weight-Based Simulation Model Construction Experiment for Leaf Area and Plant Height of Tomatoes

1. Cultivation experiment

The solar greenhouse in Experiment 1 and the continuous plastic greenhouse in Yangling SinoIsraeli Cooperation Smart Agriculture Demonstration Park, Shanxi Province (34°27′ N, 108°04′ E), were selected as the experimental areas for obtaining model validation data. The specifications of the solar greenhouse were the same as those of Experiment 2 and the tomato plant was planted on 18 October 2022, and the test tomato was 'Jinpeng No.1'; the continuous plastic greenhouse had a length of 71 m from north to south, 172 m from east to west, and a height of 8.4 m. The greenhouse had a total of 21 spans, each with a span of 8.0 m; the test tomato was 'Jinpeng No.1', and the tomato for the test was 'Golden Scaffold No.1'.

2. Validation method for the simulation of tomato plant height and leaf area based on real-time monitoring device.

In this experiment, 15 tomato plants were randomly selected in the solar greenhouse on 18 November 2022, and a total of 9 tomato plants were randomly selected in the continuous plastic greenhouse on 8 December 2022 and 21 December 2022, respectively, for the simulation and validation of tomato plant height and leaf area. First, a real-time monitoring device for tomatoes' aboveground fresh weight was set up in the greenhouse to tag the randomly selected tomato plants and measure their plant height. Afterward, the weighing device was employed to measure the aboveground weight of the labeled tomato plants for three consecutive occasions, and the mean value was considered the aboveground weight of the tomato plants. Finally, the labeled tomato plants were subjected to the destructive sampling method, and electronic scales were used to weigh the stem and leaf fresh weight of the tomato plants. The tomato's leaf area was determined using the CL-202 leaf area scanner, and the simulation model in Test 1 was used to calculate the height and leaf area of the tomato plant. A flowchart is shown in Figure 3.

3. Model validation

The root mean square error (RMSE) was selected as the index for validating the simulation accuracy of the tomato plant height and leaf area based on the real-time monitoring device.

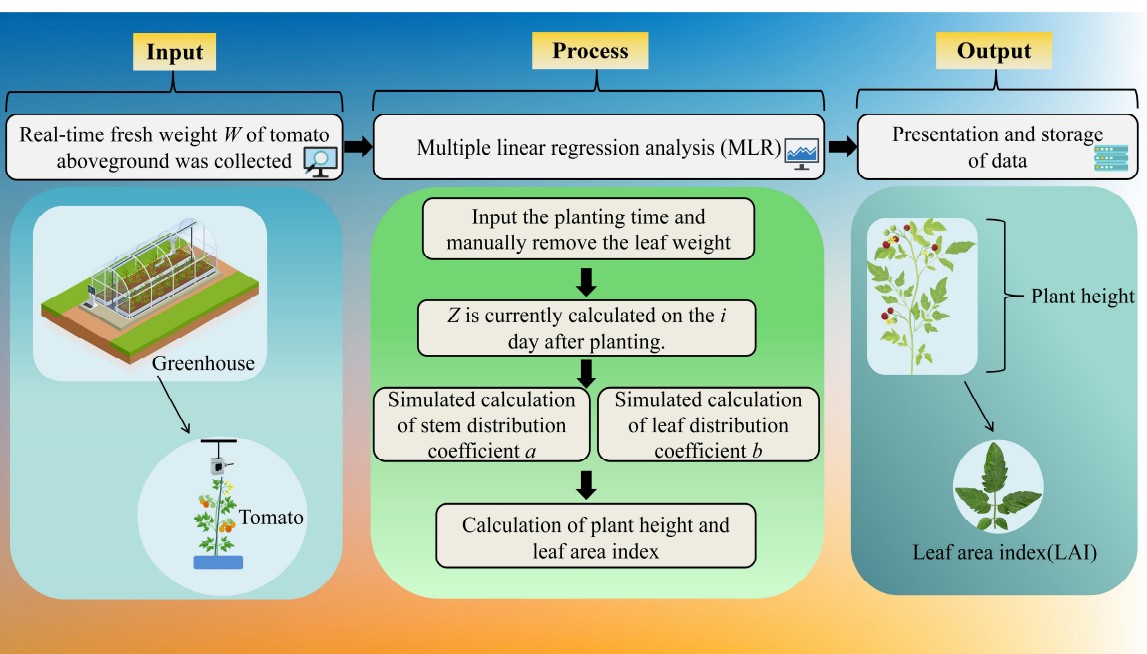

**Figure 3.** Simulation process of tomato growth index based on real-time monitoring device.

## 3. Results

### 3.1. Relationship of Tomatoes' Morphological Organs with Plant Height and Leaf Area

During Experiment 1's tomato reproductive period, we analyzed correlation coefficients between the stem and leaf fresh weight, and plant height and leaf area (Tables 1 and S1). The results showed a close link between stem weight and plant height in the spring and autumn. The correlation coefficients were very significant. The r values were 0.930 and 0.944. This relationship passed the significance test of *F* (0.01). The tomato leaves' weight was strongly linked to their size in the spring and autumn. For the spring, the correlation coefficient was 0.952, and for autumn, it was 0.984. Both coefficients passed the *F* (0.01) significance test.

**Table 1.** Correlation analysis of tomatoes' morphological organs with plant height and leaf area.

| Stubble | | | Stem Fresh Weight | | | Leaf Fresh Weight |
|---|---|---|---|---|---|---|
| Spring stubble | Plant height | $r$ | 0.930 ** | Leaf area | $r$ | 0.952 ** |
| | | $p$ | 0.001 | | $p$ | 0.001 |
| Autumn stubble | Plant height | $r$ | 0.944 ** | Leaf area | $r$ | 0.984 ** |
| | | $p$ | 0.001 | | $p$ | 0.001 |

Note: $r$ is the correlation coefficient; $p$ is significance: ** is $p < 0.01$.

### 3.2. Simulation Modeling of Tomato Plant Height and Leaf Area Index

3.2.1. Model Construction

Based on data from the 2021 insulated plastic shed fall stubble trial, Section 3.4.1 presents the 'Calculation method of tomato stem and leaf distribution coefficient', which integrates findings from previous studies and our own planting experience in this study. A methodology was developed to calculate the distribution coefficients (a) for stem organs and (b) for leaf organs in the aboveground parts of tomato plants. The total weight of aboveground tomatoes was measured using dynamometers, and by multiplying this weight (W) with the respective organ distribution coefficients (a and b), values $W_h$ and $W_l$ were obtained, respectively. Subsequently, these values were utilized along with a model to determine the height of a tomato plant (H) as well as its leaf area index (LAI). Mathematical functions were employed to construct simulation models for both the tomato plant height and leaf area index.

### 3.2.2. Model Validation

1. Plant height simulation model validation

The accuracy of the modeled tomato plant height simulations (Tables 2 and S2) was verified using measured data from spring stubble in insulated plastic greenhouses and overwintering stubble stem organs in solar greenhouses in Experiment 1, respectively.

**Table 2.** Modeling tomato morphology and appearance using different mathematical functions.

| Tomato Appearance | Math Function Types | Model |
|---|---|---|
| Plant height (H) | Linear function<br>Polynomial Function<br>Idempotent function | $H = 0.5286W_h + 26.304$<br>$H = -0.0015W_h^2 + 0.8725W_h + 17.554$<br>$H = 9.2293W_h^{0.4941}$ |
| Leaf area (LA) | Linear function<br>Polynomial Function<br>Idempotent function | $LA = 29.182W_l + 163.75$<br>$LA = -0.0917W_l^2 + 43.396W_l - 85.18$<br>$LA = 41.262W_l^{0.9445}$ |
| Leaf area index (LAI) | / | $LAI = \frac{LA}{S}$ |

Note: $W_h$ is the fresh weight of the tomato stems, $W_l$ is the fresh weight of the tomato leaves, H is the height of the tomato plants, LA is the leaf area of the tomato, S is the unit footprint of the tomato plants, which was 0.24 cm $\times$ 0.28 cm in this cultivation experiment, and LAI is the leaf area index.

Through the simulation process of the tomato plant height mentioned in Figure 3, the tomato plant height was simulated based on the three formulas of plant height, as shown in Table 1, and compared with the actual values; the comparisons of Figure 4 were obtained by means of a 1:1 line. Included among these, the tomato plant height simulation for spring crops in insulated plastic greenhouses was good. The $R^2$ values for the linear, polynomial, and power functions were 0.886, 0.888, and 0.888; all $R^2$ values were above 0.8. The RMSE values were 12.96 cm, 24.05 cm, and 12.99 cm (Figure 4a). In the simulation results of the tomato plant height for overwintering stubble in solar greenhouse, the $R^2$ values for the linear, polynomial, and power functions were 0.905, 0.919, and 0.987, with all $R^2$ values above 0.9. The RMSE values were 13.37 cm, 8.69 cm, and 14.10 cm (Figure 4b).

2. Leaf area index simulation model validation

The accuracy of the modeled tomato plant leaf area index simulations (Table 2) was verified using measured data from spring stubble in insulated plastic greenhouses and overwintering stubble leave organs in solar greenhouses in Experiment 1, respectively.

Through the simulation process of the tomato plant leaf area index mentioned in Figure 3, the tomato plant leaf area index was simulated based on the three formulas of plant height, as shown in Table 1, and compared with the actual values; the comparisons of Figure 5 were obtained by means of a 1:1 line. The results showed that in the simulation of the leaf area index of spring stubble tomatoes in insulated plastic greenhouses, the $R^2$ values of the linear, polynomial, and power index functions were 0.881, 0.843, and 0.880. All $R^2$ values were above 0.8. The RMSEs were 0.98 m$^2 \cdot$m$^{-2}$, 1.20 m$^2 \cdot$m$^{-2}$, and 1.14 m$^2 \cdot$m$^{-2}$ (Figure 5a). Meanwhile, the results show that for the simulation diagram of the leaf area index of overwintering stubble in solar greenhouse, the $R^2$ values of the linear, polynomial, and power index functions were 0.928, 0.902, and 0.928, with all $R^2$ above 0.9. The RMSEs were 0.54 m$^2 \cdot$m$^{-2}$, 0.69 m$^2 \cdot$m$^{-2}$, and 0.53 m$^2 \cdot$m$^{-2}$ (Figure 5b).

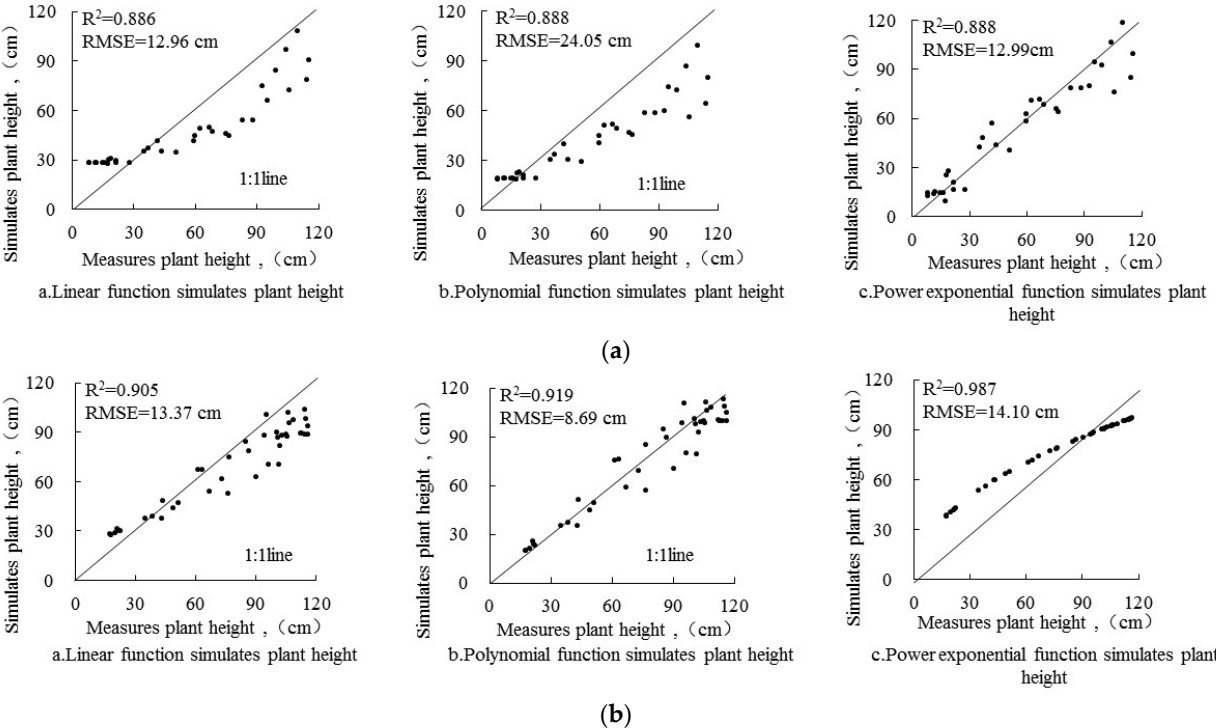

**Figure 4.** Comparison of simulated and measured tomato plant height values: (**a**) simulation of plant height in spring stubble under thermal insulation plastic greenhouse; (**b**) simulation of plant height during overwintering in solar greenhouse.

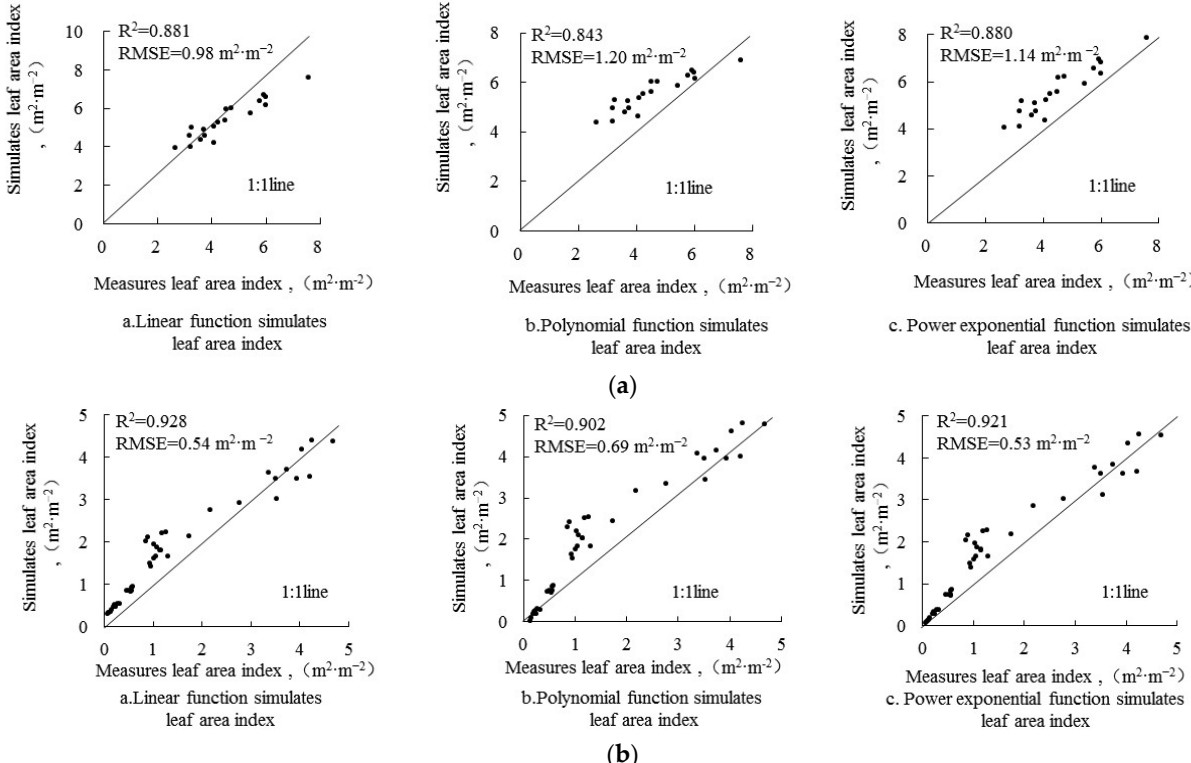

**Figure 5.** Comparison between simulated and measured values of the tomato leaf area index: (**a**) simulation diagram of the leaf area index of spring stubble in thermal insulation plastic greenhouse; (**b**) simulation diagram of the leaf area index of overwintering stubble in solar greenhouse.

### 3.3. *Tomatoes' Aboveground Fresh Weight Real-Time Monitoring Device Operation Detection*
### 3.3.1. Long-Term Monitoring of the Weight of Constant-Weight Objects

As shown in Figure 6, three sensors were used to test the stability of a tomato ground weight monitoring device. The sensors measured the weight of a constant 1.550 kg object. The S-type tension sensors collected data that showed small fluctuations, ranging from 1.55 kg up to 1.55 kg down. The monitoring weight had a maximum error of 30.81 g compared to a constant-weight object. The average error of the three groups of sensors monitoring the weight was 16.95 g. The comprehensive monitoring of the weight data of the constant-weight object was conducted for 16 consecutive d. This device can monitor the weight of suspended objects for a long time. It considers the environment, sensor voltage fluctuations, and the sensor itself. These factors affect the measurement accuracy.

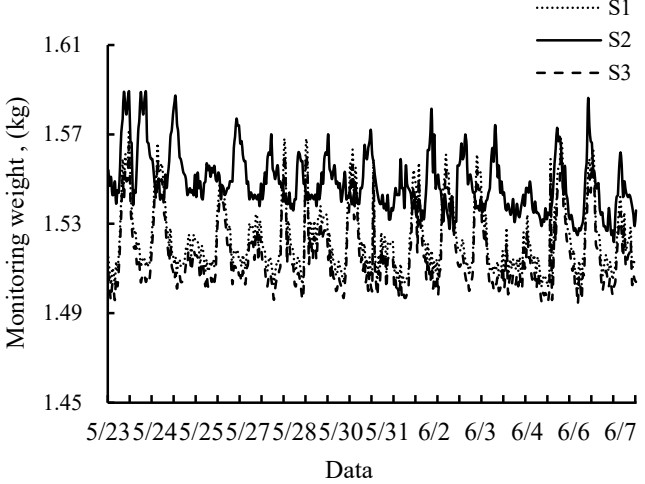

**Figure 6.** Real-time monitoring diagram of constant-weight object's weight.

### 3.3.2. Measurement Accuracy Test of Tomato Aboveground Fresh Weight Device in Different Periods

It can be seen from Figure 7 that 12 tomato plants were collected every 20 days in the solar greenhouse, for a total of 36 plants. The device measured the weight of a tomato aboveground. Then, a tool was used to cut the tomato off, and an electronic balance was used to measure its weight. Finally, the device's and electronic balance's measured values were compared. It can be seen that the fitting degree between the measured value and the measured value of the device is good; the $R^2$ was 0.937, RMSE was 0.05 kg, and MAE was 0.04 kg. The device can measure the weight of tomato plants aboveground in real time. It does not harm the average growth of tomatoes.

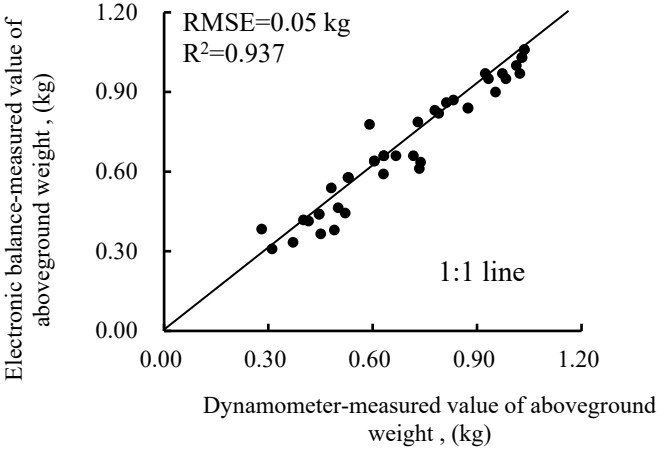

**Figure 7.** Measurement accuracy test of tomato aboveground fresh weight device.

### 3.4. Simulation of Tomato Plant Height and Leaf Area Index Model Based on Real-Time Monitoring Device

3.4.1. Calculation Method of Tomato Stem and Leaf Distribution Coefficient

In different stubbles of tomato cultivations and in tomato seedlings 10 days after planting, the slow seedling of the tomatoes ended, and they began to grow normally. The distribution index of stem organs showed a consistent and stable trend, and the distribution index of leaf organs showed a trend of slow increase at first and then decrease [27].

In previous research on the organ distribution coefficient of greenhouse tomatoes [28,29], the growth and organ distribution coefficients of the stem and leaf organs of tomato plants have been determined at different reproductive stages. Meanwhile, the allocation coefficients of stem and leaf organs at the different reproductive stages of tomato were determined by combining the actual planting of spring and fall stubbles of tomato in large-span, insulated plastic greenhouses in this study; the distribution indices of the tomato stems and leaves were fitted and analyzed, and on this basis, the change law of the distribution coefficient of both tomato stems and leaves was synthesized, and the distribution indices of the tomato stems and leaves were fitted and analyzed, and formulas for the analysis indices of the tomato stems and leaves were obtained (Figure 8).

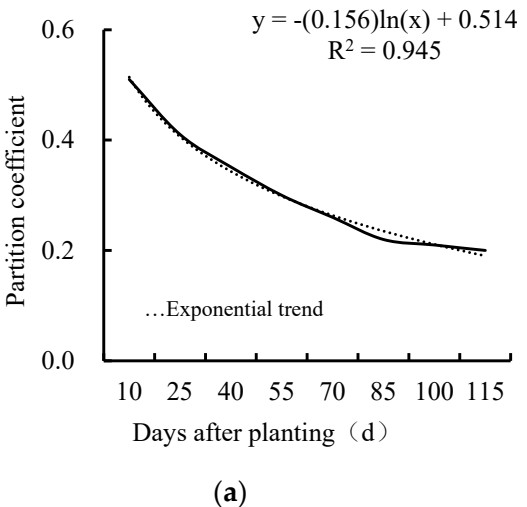
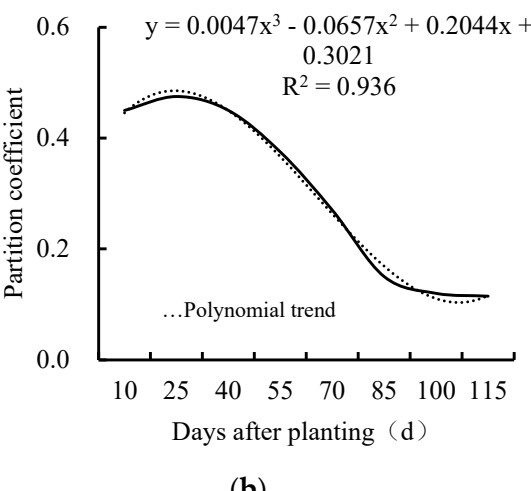

(**a**)  (**b**)

**Figure 8.** Changes in the distribution coefficient of tomato appearance and morphology: (**a**) distribution coefficient variation rule of stems; (**b**) distribution coefficient variation rule of stems.

3.4.2. Model Simulation Construction of Plant Height and LAI Based on Dynamometer

Based on the tomato growth curve and the results of Experiment 1 and the actual planting, a simulation model of the tomato plant height was created. The real-time monitoring device was used to calculate the height using linear, polynomial, and power functions (Tables 3, S3 and S4). Also, the leaf area index was calculated using the specific calculation formulas in Table 4.

**Table 3.** Plant height simulation formulas.

| Name | Model |
|---|---|
| Distribution factor | $a = -0.156\ln\left(\frac{i}{15}\right) + 0.514$ |
| Linear function | $H = 0.5286Wa + 26.304$ |
| Polynomial function | $H = -0.0015(Wa)^2 + 0.8725(Wa) + 17.552$ |
| Idempotent function | $H = 9.2293(Wa)^{0.4941}$ |

Note: i is day i after planting, a is the aboveground part of the stem allotment index, W is the aboveground weight of the tomato, and H is the height of the tomato plant.

**Table 4.** Simulation formulas for the leaf area index.

| Name | Model |
|---|---|
| Distribution factor | $b = 0.0047\left(\frac{i}{15}\right)^3 - 0.0657\left(\frac{i}{15}\right)^2 + 0.2044\frac{i}{15} + 0.3021$ |
| Linear function | $LA = 29.182(Wb + F) + 163.76$ |
| Polynomial function | $LA = -0.0917(Wb + F)^2 + 43.396(Wb + F) - 85.18$ |
| Idempotent function | $LA = 41.262(Wb + F)^{0.9445}$ |
| Leaf area index (LAI) | $LAI = \frac{LA10^{-4}}{S}$ |

Note: i is the day i after planting, b is the aboveground portion of the leaf allotment index, W is the aboveground weight of the tomato, LA is the leaf area of the tomato, S is the area occupied by a single plant of tomato (0.24 m × 0.28 m in this study), and LAI is the leaf area index of the tomato.

### 3.4.3. Application Test of Plant Height Model

In Experiment 3, in using the simulation model of tomato plant height based on real-time fresh weight, 15 tomato plants were randomly selected in the solar greenhouse, and 12 tomato plants were randomly selected in the multispan plastic greenhouse using a random sampling method, and simulation tests of tomato plant height in the two types of greenhouses were carried out using the process presented in Figure 3 to obtain the $R^2$ and RMSE values of the solar greenhouses and multispan plastic greenhouses.

The results show that in the simulation of tomato plant height in autumn in the solar greenhouse, the $R^2$ values of the linear function, polynomial function, and power exponential function were 0.813, 0.818, and 0.835, respectively, and the RMSEs were 4.46 cm, 6.71 cm, and 4.89 cm, respectively. In the simulation of the autumn tomato plant height in the multispan plastic greenhouse, the $R^2$ values were 0.858, 0.711, and 0.870, respectively, and the RMSEs were 9.24 cm, 21.86 cm, and 17.71 cm, respectively. In summary, the simulation results of the solar greenhouse are better than those of the multispan plastic greenhouse. The simulation accuracy of tomato plant height in the two greenhouses is good, and different functions can accurately simulate the tomato plant height (Table 5). The significant error between the simulated value and the measured value of the multispan plastic greenhouse results from the installation of greenhouse environment control equipment (LED fill lights, circulation fans, air source heat pump) in the multispan plastic greenhouse. The microclimate in the greenhouse is evenly distributed, which is more suitable for the growth of tomatoes, resulting in the actual value of the tomato plant height being generally more significant than the simulated value.

**Table 5.** Tomato plant height simulation based on real-time fresh weight measurement device.

| Test Greenhouse Type | Name | $R^2$ | RMSE (cm) |
|---|---|---|---|
| Solar greenhouse | Linear function | 0.813 | 4.46 |
| | Polynomial function | 0.818 | 6.71 |
| | Idempotent function | 0.835 | 4.89 |
| Multispan plastic greenhouse | Linear function | 0.858 | 9.24 |
| | Polynomial function | 0.711 | 21.86 |
| | Idempotent function | 0.870 | 17.71 |

### 3.4.4. Application Test of Leaf Area Index Model

Using the tomato leaf area index simulation model based on the real-time weights in Experiment 3, 15 and 12 tomato plants were selected in the solar greenhouse and the mul-tispan plastic greenhouse to measure the plant height. The simulated values were compared with the measured values. The simulation of the leaf area index for autumn tomatoes in the solar greenhouse had promising results. The linear, polynomial, and power exponential functions performed well. The $R^2$ values were 0.950, 0.955, and 0.953. The RMSE values were 0.20 $m^2{\cdot}m^{-2}$, 0.18 $m^2{\cdot}m^{-2}$, and 0.12 $m^2{\cdot}m^{-2}$. The leaf area index of autumn tomatoes in the plastic greenhouse was simulated. The $R^2$ values were 0.748,

0.651, and 0.870, and the RMSE values were 1.04 $m^2 \cdot m^{-2}$, 0.52 $m^2 \cdot m^{-2}$, and 1.22 $m^2 \cdot m^{-2}$ (Table 6).

**Table 6.** Tomato leaf area index (LAI) simulation based on real-time fresh weight measurement device.

| Test Greenhouse Type | Name | $R^2$ | RMSE ($m^2 \cdot m^{-2}$) |
|---|---|---|---|
| Solar greenhouse | Linear function | 0.950 | 0.20 |
| | Polynomial function | 0.955 | 0.18 |
| | Idempotent function | 0.953 | 0.12 |
| Multispan plastic greenhouse | Linear function | 0.748 | 1.04 |
| | Polynomial function | 0.651 | 0.52 |
| | Idempotent function | 0.870 | 1.22 |

## 4. Discussion

Facility microclimate plays a crucial role in tomato growth, characterized by intricate parameters and rapid fluctuations [30]. Transpiration is essential for assessing crop responses to microclimate changes caused by water stress, aiming to optimize the growing environment and crop breeding [31]. The growth and development of tomatoes are influenced by various environmental factors such as air temperature, relative humidity, and solar radiation [32,33]. To minimize simulation errors resulting from environmental conditions, this study utilized fresh weight measurements of different morphological organs in tomatoes to simulate plant height and leaf area index. Subsequently, the distribution indices of the stems and leaves were combined with the real-time monitoring of the aboveground fresh weight to establish a simulation model based on the real-time quantity of fresh tomatoes' plant height and leaf area index. This model was validated in different forks (varieties) and types of greenhouses. The simulated values for the plant height and leaf area index closely matched the measured values, indicating that the constructed simulation models have a high accuracy and applicability to traditional Chinese greenhouses (solar greenhouses and insulated plastic greenhouses). However, further optimization is required for commercially produced continuous plastic greenhouses equipped with greenhouse environmental control equipment.

In past studies on crop growth modeling, experts relied on the destructive sampling method to acquire crop growth index data [34–36]. However, this method needed to be improved upon in monitoring the same plant over an extended period, resulting in a sparse data collection and errors during model construction. This study utilized a weighing device to monitor plant growth continuously, while agricultural IoT equipment was used to visualize and analyze the data online. In a study by Liu et al. [22] on tomato appearance and morphology in insulated plastic greenhouses scaled through an irradiation heat product, it was found that the plant height had an RMSE of 13.66 cm. At the same time, the LAI was 1.03 $m^2 \cdot m^{-2}$. In this experiment, the average RMSE of the plant height was 10.81 cm, and the LAI was 0.55 $m^2 \cdot m^{-2}$. Overall, the findings indicate that the model developed in this study closely aligns with the simulation model based on irradiation heat accumulation. Notably, it requires monitoring only the fresh weight of fresh plants without necessitating the installation of environmental monitoring equipment. This feature makes it well suited for adoption in conventional greenhouses that lack sophisticated technology, enhancing its practicality.

The simulation results for the three methods were similar. All of the models simulated the height of tomato plants. The winter stubble in the solar greenhouse was more accurate than the spring stubble in the plastic greenhouse. The latter greenhouse did not have a back wall. This made it less accurate at preserving and storing heat. The second greenhouse's simulation may not handle cold temperatures well. This can cause slight inaccuracies. This can prevent tomato plants from growing well and make the simulated and real plant heights vary. The simulation accuracy of the leaf area index of overwintering tomatoes in the solar greenhouse is better than that of spring stubble in a thermal insulation plastic

greenhouse. The reason for the error may be that the solar greenhouse's thermal insulation and heat storage capacity are more robust than those of the thermal insulation plastic greenhouse. The growth of tomato leaves is inhibited under low-temperature conditions, and the leaves are prone to disease and aging. When collecting leaf area data, the diseased or aging leaves had been removed by managers through agricultural operations, resulting in an inevitable simulation error. The superposition effect between the two led to the error between the simulated and measured tomato leaf area index values. Nevertheless, it needs to be more accurate in the multispan plastic greenhouse. The accuracy difference might be because the tomato plants were still seedlings in the solar greenhouse. The simulation is more accurate because these plants were mainly growing stems and leaves.

In the actual validation, it was found that the distribution indices of the tomato morphological organs were greatly affected by the environment, and the measured values of daylight greenhouse and continuous plastic greenhouse were generally higher than the simulated values because the environments inside their chambers were more stable and more suitable for tomato growth. When long-term low-temperature, low-light, and other growth adversities occur, the dry matter allocation of tomato stems and leaves will be inhibited to different degrees, which is consistent with the results of Gao et al. [37]. The distribution coefficient significantly affects the difference between the simulated results and the measured values of the tomato plant height and leaf area index. In addition to time factors, environmental factors, water and fertilizer measures, etc., will specifically impact the tomato distribution index [38,39]. This study considered only the linear coupling between the fresh weights of tomato organs and the plant height and leaf area index, and the influence of nonlinear coupling on the final research results remains to be further investigated.

For all experiments conducted in this study, we employed the substrate bag cultivation method. In contrast, growers in traditional Chinese greenhouses mainly use soil cultivation, which can affect tomato growth differently under different cultivation conditions [40–42]. In order to improve the applicability of the model in traditional Chinese greenhouses, the model will be validated in the future in different types of greenhouses, using different cultivation methods, and in different climatic zones, and based on this, the model will contribute to the future development of tomato-related research on water demand and irrigation decision-making devices.

## 5. Conclusions

The study suggests a new way for monitoring the greenhouse tomato plant height and leaf area index. This method is nondestructive and based on real-time simulation. It offers a fresh approach to building greenhouse tomato simulations. It also helps estimate tomato crop transpiration and create water-saving irrigation strategies as follows:

(1) In this study, we developed and installed a long-term, real-time monitoring device for tomato aboveground wire volume, which monitors the aboveground weight of tomato plants in real time without disrupting the average growth of tomatoes, with a coefficient of determination $R^2$ of 0.937, RMSE of 0.05 kg, and MAE of 0.04 kg;

(2) Simulation models of the tomato plant height and leaf area index based on real-time weight were constructed, through which the predicted values of the tomato plant height and leaf area index in different greenhouses were estimated to fit well with the measured values. The average coefficient of determination $R^2$ in the simulation of the plant height was 0.817, the RMSE was 10.81 cm, and the integrated simulation effect of the linear function was good; the average coefficient of determination $R^2$ in the simulation of the leaf area index was 0.854, the RMSE was 0.55 $m^2 \cdot m^{-2}$, and the polynomial function simulation was better. The model can be used to estimate the plant height and leaf area index in real time, which provides a new way of thinking for the construction of a simulation model of tomato growth indices in facilities and also establishes a particular foundation for the estimation of tomato evapotranspiration and the formulation of water-saving irrigation strategies.

**Supplementary Materials:** The following supporting information can be downloaded at https://www.mdpi.com/article/10.3390/horticulturae10030270/s1: Table S1 Correlation analysis of tomatoes' morphological organs with plant height and leaf area; Table S2 Model construction and model validation data; Table S3 Accuracy testing of dynamometer devices; Table S4 Dynamometer-based simulation model testing of plant height and leaf area index.

**Author Contributions:** Data collection, S.G. and L.W.; data curation, S.G.; figures, L.W.; formal analysis, Y.L. and X.S.; supervision, H.S. and Y.C.; resources, Y.C.; writing—review and editing, S.G., L.W., X.C., X.S., Y.C., Y.L. and H.S.; writing instruction, Y.C. and H.S. All authors have read and agreed to the published version of the manuscript.

**Funding:** This research was funded by the Xinjiang Uygur Autonomous Region Major Sciences and Technology Special Project (2022A02005-5, 2022A02005-1), Xinjiang Uygur Autonomous Region Public Welfare Scientific Research Institutes Project (KY2023036), and Key Laboratory: Xinjiang Key Laboratory of Intelligent Agricultural Facility Management and Control Technology (XJYS1703).

**Data Availability Statement:** The original contributions presented in the study are included in the article and supplementary material, further inquiries can be directed to the corresponding authors.

**Acknowledgments:** We fully appreciate the editors and all anonymous reviewers for their constructive comments on this manuscript.

**Conflicts of Interest:** Author Letian Wu, Xinwei Cao, Xiaoli Sun, and Huifen Shi were employed by the Agricultural Engineering Company. The remaining authors declare that the research was conducted in the absence of any commercial or financial relationships that could be construed as a potential conflict of interest.

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
