# Peer review of "Simulation Model Construction of Plant Height and Leaf Area Index Based on the Overground Weight of Greenhouse Tomato: Device Development and Application"

_horticulturae, doi:10.3390/horticulturae10030270_

Round 1

Reviewer 1 Report

Comments and Suggestions for Authors

This study was conducted to establish a mathematical simulation model of tomato leaf area index and plant height based on real-time fresh weight. The idea of work is interesting, there are some comments:

The structure of the abstract could be better, the framework of the abstract section include a brief introduction, material and method, results, and an impressive conclusion. In your research lines 14-20 are the introduction, lines 20-22 are the results, lines 23-25 are the method, lines 26-27 are the results, and ....also the conclusion is not impressive. Please rewrite this section.

Line 36: ev-er --> ever

Lines 40-42: This part is not appropriate with your research, please remove or replace this sentences with scientific states!!! Before the covid pandemic and ... there were many problems in some part of the world, specially in some African countries. The authors should bring general and scientific states.

Here is a newly published work on tomato growth in greenhouse and farm that fit with your scope. I would suggest that the authors review and include this study to improve the introduction https://doi.org/10.3390/agronomy13030916 

Line 139: ... conventional field management --> please add some details in regard with conventional field management such as the type of agricultural operation, agricultural inputs (fertilizers and ...).

Lines 145-146: For height measurement, every 7 days is a short period and usually measurements are performed two times per months.

The irrigation is a main part of your research and in the material and method section this part is missing, please add some details.

The result is well described.

The discussion is good, but can be better with more expansion and compare with other research and highlight the novelty of current research. 

Line 597: This reference is repetitive. Please remove it. 

Comments on the Quality of English Language

Minor editing of English language required

Author Response

Dear reviewer:

Thank you very much for giving us an opportunity to revise our manuscript. The authors would like to acknowledge the encouraging and valuable comments concerning our manuscript entitled “Simulation model construction of plant height and leaf area index based on the overground weight of greenhouse tomato and device development and application”(ID: horticulturae-2757875 ). Those comments are all valuable and very helpful for revising and improving our paper, as well as the important guiding significance to our researches. We have carefully studied the relevant comments and have made major revision on the manuscript for your kind consideration for publication. Revised portion are highlighted in yellow in the manuscript. The main corrections in the present work and the responds to the reviewer’s comments are as follows:

Responds to the Academic Editors comments:

Points in Favor:

This study was conducted to establish a mathematical simulation model of tomato leaf area index and plant height based on real-time fresh weight. The idea of work is interesting.

Response:

Thank you sincerely for your invaluable suggestions on our manuscript. My co-authors and I have been diligently engaged in the scientific investigation of greenhouse environments and crop growth models. We would like to extend our utmost appreciation to the editor and reviewers for their valuable input. In accordance with the reviewers' comments, we have meticulously revised every aspect of the manuscript.

Points Detracting:

Comments 1:

The structure of the abstract could be better, the framework of the abstract section include a brief introduction, material and method, results, and an impressive conclusion. In your research lines 14-20 are the introduction, lines 20-22 are the results, lines 23-25 are the method, lines 26-27 are the results, and ....also the conclusion is not impressive. Please rewrite this section.

Response1:

Thank you very much for your kind work and valuable suggestions. The abstract is an important part of the paper that summarizes the whole text, removes loose ends and extracts the main information of the paper. We will rewrite the abstract according to your requirements.

Comments 2:

Line 36: ev-er --> ever

Lines 40-42: This part is not appropriate with your research, please remove or replace this sentences with scientific states!!! Before the covid pandemic and ... there were many problems in some part of the world, specially in some African countries. The authors should bring general and scientific states.

Here is a newly published work on tomato growth in greenhouse and farm that fit with your scope. I would suggest that the authors review and include this study to improve the introduction https://doi.org/10.3390/agronomy13030916

Response2:

Thank you for raising these points. As the commenter suggested:

We changed the formatting of line 36.

We have deleted the introductory section in lines 40-42, and we have given much thought to your suggestion that we should make general and scientific statements.

As we all know that greenhouse tomato production is an important part of facility agriculture, we have taken your advice and downloaded and carefully read "Impacts of Environmental Factors and Nutrients Management on Tomato Grown under Controlled and Open Field Conditions" and referenced it for greenhouse and on-farm tomato production, thank you again for your advice!

Comments 3:

Line 139: ... conventional field management --> please add some details in regard with conventional field management such as the type of agricultural operation, agricultural inputs (fertilizers and ...).

Lines 145-146: For height measurement, every 7 days is a short period and usually measurements are performed two times per months.

Response3:

Thank you for your suggestion. It is important to make sure that the article is rigorous.This paper describes conventional field management, where different field management results in different tomato growth outcomes. The article focuses on mathematical modeling, so field management defaults to what most people do in conducting greenhouse tomato production.This paper establish a mathematical simulation model of tomato leaf area index and plant height based on real-time fresh weight.In order to make the model universally applicable and to be able to be applied directly in Chinese traditional greenhouse tomato production。 In order to make the model directly applicable in traditional Chinese greenhouse tomato production, farmers with many years of growing experience were sought for the experimental greenhouse management, and their methods covered most of the management methods of most farmers with rich growing experience for different stages of greenhouse tomato growth. Therefore considered to be generally the same as the management methods of popular growers by default.

To improve the mathematical model's simulation accuracy, plant height measurements were taken every 7 days; the shorter the interval, the more accurate tomato growth monitoring. Therefore, this article uses 7 days as the collection interval for plant height.

Comments 4:

The irrigation is a main part of your research and in the material and method section this part is missing, please add some details.

The result is well described.

The discussion is good, but can be better with more expansion and compare with other research and highlight the novelty of current research.

Line 597: This reference is repetitive. Please remove it.

Response4:

According to the reviewer’s comments, we have revised these descriptions:

In this paper, tomatoes are grown in substrate bag culture, and irrigation is managed by farmers with many years of experience in growing tomatoes, usually according to plant morphology, external climate and other conditions. Generally 5 days after planting watering 1, to keep the inter-root substrate moist, anti-growth.  Watering once a day on a sunny day. Depending on the specific circumstances of cloudy days, less watering or no watering. Irrigation was described in Materials and Methods,this method is universally representative of traditional greenhouse production in China.

We have added to the discussion by highlighting the novelty of current research and thank you for your valuable suggestions!

We have removed the duplicate reference at line 597.

Reviewer 2 Report

Comments and Suggestions for Authors

The article theme and article is very interesting and the experiments are well conducted. The article:

Simulation model construction of plant height and leaf area index based on the overground weight of greenhouse tomato and device development and application.

has been reviewed obtaining the following observations:

Check the word ev-er in line 39.

Line 73 change word to in-crease, simu-lation in line 74, de-veloped in line 76 and in-sulated in line 77. There are more in the introduction section.

Check line 116.

I do not find any difference between figure 1 and figure 2 solar GH. Please highlight it in the figure.

In line 144 eliminate dry and fresh.

Change tape measure in line 147.

Please rewrite paragraph from line 190 to 194 so that it can be understood easily.

The point in the heading of Figure 3 should have the point after the 3.

AFTER READING THE METHODOLOGY SECTION I BELIEVE THAT THE EXPERIMENTAL SETUP SHOULD BE CLEARLY EXPLAINED AT THE BEGINNING OF SECTION 2 INDICATING THAT THERE ARE 3 EXPERIMENTAL GREENHOUSES.

Repeat lines 251 and 252 so that they mean what the regression values show in Table 1.

In line 254 and 258 and Table 1 the regression variable is R2.

What is the meaning of mold in Table 2. Also please explain what you mean by Idempotent.

What you mean by stubble?

Paragraph 282-288 should be changed to discussion section and eliminate in summary.

In line 302 add depending after dataset.

Please rewrite lines 305-309 to be clearer. Divide it in 2 shorter sentences.

The paragraph between lines 3122-320 should be moved to the discussion section.

Figure 6a should appear in text. It should be in line 280 and 279. Figure 6b in line 276. Change stubble by experiment in the heading of Figure 6 and eliminate “/“ in all the vertical axis of both Figures 6 and 7 and put a “,”.  

Figure 7a should appear on line 307 and Figure 7b on line 309. You repeated twice the values of R2 in this line.

Line 331 is not correct: ranging from 1.55kg 335 up to 1.55kg down. It only had a a deviation of 31 g.

In figure 9 both axis have the same measurement.

Eliminate in this study in line 365.

Figure 10 does not appear in the text and there is no explanation of it.

Eliminate created in line 373 and add was created after plant height at the end of the sentence.

Change the note below Table 3 to line 375 after (Table 3). Also change the note below Table 4 to line 376.

Heading 3.4.2 presents an italics font in line 370.

There are missing points and spaces in Table 3, Table 4, Table 5 headings and also in heading 3.4.5.

It is missing a space between 858 and 711 in line 392. The value of 0.711 cannot be considered a good accuracy.

In line 395 change places by greenhouses.

Table 5 does not appear anywhere in the text. I think it should be added in line 390 and 392.

In line 422 you put table 6 that does not exist.

Paragraph from lines 414 to 422 must be changed to the discussion section.

In line 441 eliminate often.

IN THE ARTICLE YOU TALK ABOUT LAI but where is relationship between leaf area and planting area considering all the levels of the plant.

REFERENCES

Between the journal name and the publication year there is no point.

Author Response

Dear reviewer:

Thank you very much for giving us an opportunity to revise our manuscript. The authors would like to acknowledge the encouraging and valuable comments concerning our manuscript entitled “ Simulation model construction of plant height and leaf area index based on the overground weight of greenhouse tomato and device development and application” (ID: horticulturae-2757875 ). Those comments are all valuable and very helpful for revising and improving our paper, as well as the important guiding significance to our researches. We have carefully studied the relevant comments and have made major revision on the manuscript for your kind consideration for publication. Revised portion are highlighted in yellow in the manuscript. The main corrections in the present work and the responds to the reviewer’s comments are as follows:

Responds to the Academic Editors comments:

Points in Favor:

The article theme and article is very interesting and the experiments are well conducted.

Response:

Thank you sincerely for your invaluable suggestions on our manuscript. My co-authors and I have been diligently engaged in the scientific investigation of greenhouse environments and crop growth models. We would like to extend our utmost appreciation to the editor and reviewers for their valuable input. In accordance with the reviewers' comments, we have meticulously revised every aspect of the manuscript.

Points Detracting:

Comments 1:

Check the word ev-er in line 39.

Line 73 change word to in-crease, simu-lation in line 74, de-veloped in line 76 and in-sulated in line 77. There are more in the introduction section.

Check line 116.

I do not find any difference between figure 1 and figure 2 solar GH. Please highlight it in the figure.

In line 144 eliminate dry and fresh.

Change tape measure in line 147

Please rewrite paragraph from line 190 to 194 so that it can be understood easily.

Response1:

Thank you very much for your kind work and valuable suggestions. We have checked the whole text and corrected similar formatting errors such as "check the word ev-er in line 39".

We checked 116 line. Thank you for your attention. Due to an error in uploading the manuscript earlier, the diagrams of the greenhouse structure in Figures 1 and 2 were misrepresented as the same diagram. For clarification, Figure 1 is a schematic diagram of a large-span asymmetrical plastic greenhouse, while Figure 2 is a structural diagram of a solar greenhouse. In addition, Figure 4 is a schematic of the structure of a continuous plastic greenhouse, which actually represents a different greenhouse structure.

We have eliminated dry and fresh in line 144.

We have re-described the measurement of tomato plant height in line 147. Tomato plant height is measured by the conventional method of using a straightedge to measure the distance from the bottom of the tomato to the top growing point as the tomato plant height.

To make it easier to understand, we have rewritten the paragraph from line 190 to line 194:Tomato growth information collection control cabinet is based on the Internet of Things (IoT) technology., which includes an acquisition microcontroller (DAM3232, China Jiuying Soaring Electronics), a data 4G network transmission module, and an MCGS display, and the data was collected into the acquisition microcontroller through dynamometers, and the MCGS display was used locally as a monitoring module for the grower to view the tomato growth data in the greenhouse in real time , and then it is uploaded to the cloud through the 4G network transmission module, and the schematic diagram of information collection is shown in Fig. 3.

Comments 2:

The point in the heading of Figure 3 should have the point after the 3.

AFTER READING THE METHODOLOGY SECTION I BELIEVE THAT THE EXPERIMENTAL SETUP SHOULD BE CLEARLY EXPLAINED AT THE BEGINNING OF SECTION 2 INDICATING THAT THERE ARE 3 EXPERIMENTAL GREENHOUSES.

Repeat lines 251 and 252 so that they mean what the regression values show in Table 1.

In line 254 and 258 and Table 1 the regression variable is R2.

Response2:

Thank you for raising these points. As the commenter suggested:

We have changed the title of Figure 3, thank you for your careful labeling!

We have redescribed the setup of the experiments in this paper in Section 2, noting that there are three greenhouses (Fig. 1, Fig. 2, Fig. 4), Fig. 1 represents the Schematic diagram of a large-span asymmetric plastic greenhouse, Fig. 2 represents the Structural diagram of a solar greenhouse. Fig. 4 represents Structural schematic diagram of a continuous plastic greenhouse.

We've made changes to lines 251 and 252.

We have carefully read your proposed changes and reviewed the literature, and the meaning we want to convey here is: "Correlation analysis of tomato morphological organs with plant height and leaf area". The correlation coefficient r and R2 are two indicators commonly used in statistics to measure the strength of the linear relationship between two variables. Here r is Pearson's correlation coefficient ,it is the result of correlation analysis of tomato plants (1. plant height and total weight of stem organs; 2. leaf area and total weight of leaf organs) using SPSS software.

Comments 3:

What is the meaning of mold in Table 2. Also please explain what you mean by Idempotent.

What you mean by stubble?

Paragraph 282-288 should be changed to discussion section and eliminate in summary.

In line 302 add depending after dataset.

Please rewrite lines 305-309 to be clearer. Divide it in 2 shorter sentences.

The paragraph between lines 312-320 should be moved to the discussion section.

Response3:

According to the reviewer’s comments, we have revised these descriptions:

“Mold” in Table 2 means calculated formula, which has now been modified to formula for ease of reading and understanding. “Idempotent” means idempotent function (math.),it is a functional differentiation method.

Tomatoes are grown in several seasons throughout the year, let's say spring stubble is planted in spring, the reference to spring stubble in this article may be a direct translation of the Chinese, which is now modified to spring crop and autumo crop, respectively.

We have changed lines 282-288 to the discussion section, thank you for your valuable suggestion, we do realize that something is wrong!

We have finished adding "depending" on line 302.

We have revised lines 305-309 in the original text into 2 short sentences to make them concise and clear to understand: The results showed that in the simulation of the leaf area index of spring-crop tomatoes in insulated plastic greenhouses, the R2 of linear, polynomial, and power-index functions were 0.881, 0.843, and 0.880. The RMSEs were 0.98m2·m-2, 1.20m2·m-2, and 1.14m2·m-2, The results showed that Simulation diagram of leaf area index of overwintering stubble in solar greenhouse, the R2 of linear, polynomial, and power-index functions were 0.928, 0.902, and 0.928, 0.902, and 0.928, respectively. Index function, the R2 of linear function, polynomial function, and power function were 0.928, 0.902, and 0.921, respectively, and the RMSE was 0.54m2·m-2, 0.69m2·m-2, and 0.53m2·m-2.

Comments 4:

Figure 6a should appear in text. It should be in line 280 and 279. Figure 6b in line 276. Change stubble by experiment in the heading of Figure 6 and eliminate “/ “in all the vertical axis of both Figures 6 and 7 and put a “,”. 

Figure 7a should appear on line 307 and Figure 7b on line 309. You repeated twice the values of R2 in this line.

Line 331 is not correct: ranging from 1.55kg 335 up to 1.55kg down. It only had a a deviation of 31 g.

Response4:

Thank you for your suggestion. It is important to make sure that the article is rigorous. We have placed figures 6a and 6b in the article, and changed "stubble" to "experiment" in the title of figure 6, and eliminate “/ “in all the vertical axis of both Figures 6 and 7 and put a “,”.

The results showed that in the simulation of the leaf area index of spring-crop tomatoes in insulated plastic greenhouses, the R2 of linear, polynomial, and power-index functions were 0.881, 0.843, and 0.880. The RMSEs were 0.98m2·m-2, 1.20m2·m-2, and 1.14m2·m-2, respectively. The results showed that in the simulation of the leaf area index of spring-crop tomatoes in solar greenhouses, the R2 of linear, polynomial, and power-index functions were 0.928, 0.902, and 0.928, 0.902, and 0.928, respectively. Index function, the R2 of linear function, polynomial function, and power function were 0.928, 0.902, and 0.921, respectively, and the RMSE was 0.54m2·m-2, 0.69m2·m-2, and 0.53m2·m-2, respectively.

Comments 5:

In figure 9 both axis have the same measurement.

Eliminate in this study in line 365.

Figure 10 does not appear in the text and there is no explanation of it.

Eliminate created in line 373 and add was created after plant height at the end of the sentence.

Change the note below Table 3 to line 375 after (Table 3). Also change the note below Table 4 to line 376.

Heading 3.4.2 presents an italics font in line 370.

Response5:

Based on what you suggested, we have made the following changes:

In figure9,this graph shows the "Tomato Above Ground Fresh Weight Device Measurement Accuracy Test" where accuracy is the weight measured by the device compared to the weight of tomato plants weighed on an electronic scale after destructive sampling and plotted on a "1:1 line" so that both axes have the same measurement.As can be seen from the 1:1Line, the device has a high degree of accuracy with an RMSE of 0.05 kg and an R2 of 0.937.

We've deleted the contents of line 365.

We have already cited and described Figure 10 already in Section 3.4.1.

We have removed "plant height" from line 373 and added "add" after "plant height" at the end of the sentence.

We have Change the note below Table 3 to line 375 after (Table 3). Also change the note below Table 4 to line 376.Also italicize line 370 heading 3.4.2

Comments 6:

There are missing points and spaces in Table 3, Table 4, Table 5 headings and also in heading 3.4.5.

It is missing a space between 858 and 711 in line 392. The value of 0.711 cannot be considered a good accuracy.

In line 395 change places by greenhouses.

Table 5 does not appear anywhere in the text. I think it should be added in line 390 and 392.

In line 422 you put table 6 that does not exist.

Paragraph from lines 414 to 422 must be changed to the discussion section.

In line 441 eliminate often.

Response6:

We have made changes to the formatting of the headings and other elements of Tables 3, 4 and 5.

A space is missing between 858 and 711 in line 392. We have added it, thank you for your careful observation!

The value of 0.711 cannot be considered a good accuracy.

“The value of 0.711 cannot be considered a good accuracy.”We have already analyzed it in our discussion, due to the more frequent management of commercial greenhouses, the pruning of old leaves and the use of environmental control equipment such as air source heat pumps and circulating fans in the greenhouses, which resulted in a large error of only 0.711.

“In line 395 change places by greenhouses.”We have modified and replaced.

“Table 5 does not appear anywhere in the text. I think it should be added in line 390 and 392.”We have added Table 5 between lines 390 and 392.

We have changed lines 414t o 422 to a discussion section and removed "often" from line 441.

Comments 7:

IN THE ARTICLE YOU TALK ABOUT LAI but where is relationship between leaf area and planting area considering all the levels of the plant.

REFERENCES

Between the journal name and the publication year there is no point.

Response7:

Thank you for your kind work and valuable suggestions.The normal growth of a tomato plant is divided into an underground part (the root system) and an above-ground part (the stem, leaves, flowers and fruits).

W is the total weight of the tomato plant, Wh is the total weight of the stem organs of the above-ground part, Wl is the total weight of the leaf organs of the above-ground part, and S is the area of the tomato plant per unit (planting density), which was calculated from the actual measurement of the tomato planting area in this article. This article describes the "Calculation method of tomato stem and leaf distribution coefficient" in Section 3.4.1, combining the results of previous studies and the actual planting experience of this article, we established a method to calculate the distribution coefficients of stem organs (a) and leaf organs (b) in the above-ground part of tomato. The total weight of the tomato above ground was determined by dynamometers, and the total weight (W) was multiplied by the distribution coefficients of the organs (a and b) to obtain Wh and Wl, respectively, and then the height of the tomato plant (H) and the leaf area index (LAI) were calculated using the model.

According to Table 4, the formula for calculating the leaf area index LAI, S is the area of the tomato plant per unit (planting density), which was calculated from the actual measurement of the tomato To improve the general applicability of the mathematical model in traditional Chinese greenhouse tomato production, the leaf area of the tomato as a whole was used for the calculation of LAI in this article.

We have removed the "point" between the journal name and the year of publication.

Round 2

Reviewer 2 Report

Comments and Suggestions for Authors

The new peer review of the article:

Simulation model construction of plant height and leaf area in- 2 dex based on the overground weight of greenhouse tomato and 3 device development and application

And the following comments are done.

The abstract is too long. I would eliminate lines 29-38. Pass them to the conclusion.

Eliminate parenthesis in line 124 and the point after figure. In line 124 and 125.

Eliminate parenthesis in line 126.

There are many words with a dash

Figure 4 comes in the text before figure 3. They must be un numerical order.

In the note of line 281 r and p should be in italics. P is not capital letter.

In Table 2, the bold title within the Table is not correct. Change mold by model.

It is not clear in Fig 6 a (line 314) what are the three top images and the 3 bottom images. At the end there are 6 images and none are explained in the text clearly.

The same occurs with Fig 6 b in page 10.

The same problem is noted with figure 7 in page 11.

Eliminate construction in Title 3.4.2 line 381.

Why Figure 9 have the same text in the axis? I Asked in the previous evaluation and it remains the same.

Figure 10 a and 10 b are not explained and should be explained clearer.

In table 3 and 4 the mold error is again present it should be model.

The new added paragraph in section 3.4.3 is not clear.

A discussion doesn’t start with “In summary” as line 459.

Comments on the Quality of English Language

There are many dashes.

Author Response

Dear reviewer2:

Thank you once again for giving me a chance to revise my manuscript. It is now during the Chinese New Year, a traditional festival in China, first of all, I wish you a happy new year, good health and all the best! Thereviewers would like to acknowledge the encouraging and valuable comments concerning our manuscript entitled “Simulation model construction of plant height and leaf area index based on the overground weight of greenhouse tomato and device development and application” (ID: horticulturae-2757875). Those comments are all valuable and very helpful for revising and improving our paper, as well as the important guiding significance to our researches. We have carefully studied the relevant comments and have made major revision on the manuscript for your kind consideration for publication. Revised portion are highlighted in yellow in the manuscript. The main corrections in the present work and the responds to the reviewer’s comments are as follows:

Responds to the Reviewer2’s comments:

Points Detracting:

Comments 1:

The abstract is too long. I would eliminate lines 29-38. Pass them to the conclusion.

Eliminate parenthesis in line 124 and the point after figure. In line 124 and 125.

Eliminate parenthesis in line 126.

There are many words with a dash.

Response1:

According to your valuable suggestions, we deleted lines 29-38 from the abstract and placed it in the conclusion and revised the abstract as a whole.

We eliminated parenthesis in line 124 and the point after figure. In line 124 and 125.

We eliminated parenthesis in line 126.

We checked the full text and removed the redundant dashs.

Comments 2:

Figure 4 comes in the text before figure 3. They must be un numerical order.

In the note of line 281 r and p should be in italics. P is not capital letter.

In Table 2, the bold title within the Table is not correct. Change mold by model.

It is not clear in Fig 6 a (line 314) what are the three top images and the 3 bottom images. At the end there are 6 images and none are explained in the text clearly.

The same occurs with Fig 6 b in page 10.

The same problem is noted with figure 7 in page 11.

Eliminate construction in Title 3.4.2 line 381.

Response2:

It is really true as your suggestion. Figure 4 is an illustration of the "Real time weight based simulation model construction experiment for leaf area and plant height in tomato" that appears in Section 2.1. It is introducing the three experiments in this paper, and therefore Figure 4 is positioned before Figure 3 in the article.

We made case changes to r and p in line 281, and made changes to the formatting of the headings in Table 2. We also modify the results and analysis of the plots in Figures 6 and 7 in more detail. We eliminated construction in Title 3.4.2 line 381.

Comments 3:

Why Figure 9 have the same text in the axis? I Asked in the previous evaluation and it remains the same.

Figure 10 a and 10 b are not explained and should be explained clearer.

In table 3 and 4 the mold error is again present it should be model.

The new added paragraph in section 3.4.3 is not clear.

A discussion doesn’t start with “In summary” as line 459.

Response3:

Thank you for bringing this issue to our attention. In accordance with your suggestion, we modified the axes of Fig. 9 to show the values of tomato plant weight measured using an electronic balance and the values of tomato plant weight measured using a Dynamometer.

We have rewritten Figures 10a and 10b. Figures 10a and 10b depict the method of calculating the weights of above-ground stem organs and leaf organs of tomatoes, respectively, which is based on a combination of our actual cultivation and the results of previous experiments in greenhouses in China.

We have amended the headings of Tables 3 and 4.

We have amended section 3.4.3 to make it clearer.

We removed the “In summary” at line 459 and optimized the discussion section.

Round 3

Reviewer 2 Report

Comments and Suggestions for Authors

Evaluation of article:
Simulation model construction of plant height and leaf area index based on the overground weight of greenhouse tomato and device development and application
Final considerations: It should be accepted.
The abstract has a right length.
Structural and schematic has the initial capital letter in lines 112, 113 and 114.
Change figure 4 to figure 3 so that they follow an order both within the text and figure subheading.
The same will occur for Figure 3 that should be changed to figure 4.
Figure 9 is easily understood now.
Tables 2mand 3 are correct.
Maybe the in figure 10 the crossing references in the x Axis should coincide with the numbers.